# EMPIRICAL BOUNDS ON LINEAR REGIONS OF DEEP RECTIFIER NETWORKS

## ABSTRACT

One form of characterizing the expressiveness of a piecewise linear neural network is by the number of linear regions, or pieces, of the function modeled. We have observed substantial progress in this topic through lower and upper bounds on the maximum number of linear regions and a counting procedure. However, these bounds only account for the dimensions of the network and the exact counting may take a prohibitive amount of time, therefore making it infeasible to benchmark the expressiveness of networks. In this work, we approximate the number of linear regions of specific rectifier networks with an algorithm for probabilistic lower bounds of mixed-integer linear sets. In addition, we present a tighter upper bound that leverages network coefficients. We test both on trained networks. The algorithm for probabilistic lower bounds is several orders of magnitude faster than exact counting and the values reach similar orders of magnitude, hence making our approach a viable method to compare the expressiveness of such networks. The refined upper bound is particularly stronger on networks with narrow layers.

## 1 INTRODUCTION

Neural networks with piecewise linear activations have become increasingly more common along the past decade, in particular since Nair & Hinton (2010) and Glorot et al. (2011). The simplest and most commonly used among such forms of activation is the Rectifier Linear Unit (ReLU), which outputs the maximum between 0 and its input argument (Hahnloser et al., 2000; LeCun et al., 2015). In the functions modeled by these networks, we can associate each part of the domain in which the network corresponds to an affine function with a particular set of units having positive outputs. We say that those are the active units for that part of the domain. Counting these "pieces" into which the domain is split, which are often denoted as linear regions or decision regions, is one way to compare the expressiveness of models defined by networks with different configurations or coefficients. The theoretical analysis of the number of input regions in deep learning dates back to at least Bengio (2009), and more recently Serra et al. (2018) have shown empirical evidence that the accuracy of similar rectifier networks can be associated with the number of such regions.

From the study of how many linear regions can be defined on such a rectifier network with $n$ ReLUs, we already know that not all configurations – and in some cases none – can reach the ceiling of $2^n$ regions. We have learned that the number of regions may depend on the dimension of the input as well as on the number of layers and how the units are distributed among these layers. On the one hand, it is possible to obtain neural networks where the number of regions is exponential on network depth (Pascanu et al., 2014; Montúfar et al., 2014). On the other hand, there is a bottleneck effect by which the width of each layer affects how the regions are partitioned by subsequent layers due to the dimension of the space containing the image of the function, up to the point that shallow networks define the largest number of linear regions if the input dimension exceeds $n$ (Serra et al., 2018).

The literature on this topic has mainly focused on bounding the maximum number of linear regions. Lower bounds are obtained by constructing networks defining increasingly larger number of linear regions (Pascanu et al., 2014; Montúfar et al., 2014; Arora et al., 2018; Serra et al., 2018). Upper bounds are proven using the theory of hyperplane arrangements by Zaslavsky (1975) along with other analytical insights (Raghu et al., 2017; Montúfar, 2017; Serra et al., 2018). These bounds are only identical – and thus tight – in the case of one-dimensional inputs (Serra et al., 2018). Both of these lines have explored deepening connections with polyhedral theory, but some of these

results have also been recently revisited using tropical algebra (Zhang et al., 2018; Charisopoulos & Maragos, 2018). In addition, Serra et al. (2018) have shown that the linear regions of a trained network correspond to a set of projected solutions of a Mixed-Integer Linear Program (MILP).

Other methods to study neural network expressiveness include universal approximation theory (Cybenko, 1989), VC dimension (Bartlett et al., 1998), and trajectory length (Raghu et al., 2017). Different networks can be compared by transforming one network to another with different number of layers or activation functions. For example, it has been shown that any continuous function can be modeled using a single hidden layer of sigmoid activation functions (Cybenko, 1989). In the context of ReLUs, Lin & Jegelka (2018) have shown that the popular ResNet architecture (He et al., 2016) with a single ReLU neuron in every hidden layer can be a universal approximator. Furthermore, Arora et al. (2018) have shown that a network with single hidden layer of ReLUs can be trained for global optimality with a runtime polynomial in the data size, but exponential in the input dimension. The use of trajectory length for expressiveness is related to linear regions, i.e., by changing the input along a one dimensional path we study the transition in the linear regions.

Certain critical network architectures using leaky ReLUs ($f(x) = max(x, \alpha x), \alpha \in (0, 1)$) are identified to produce connected decision regions (Nguyen et al., 2018). In order to avoid such degenerate cases, we need to use sufficiently wide hidden layers. However, this result is mainly applicable for leaky ReLUs and not for the standard ReLUs (Beise et al., 2018).

Although the number of linear regions has been long conjectured and recently shown to work for comparing similar networks, this metric would only be used in practice if we come up with faster methods to count or reasonably approximate such number. Our approach in this paper consists of introducing empirical upper and lower bounds, both of which based on the weight and bias coefficients of the networks, and thus able to compare networks having the same configuration of layers.

In particular, we reframe the problem of determining the potential number of linear regions $N$ of an architecture with that of estimating the *representation efficiency* $\eta = \log_2 N$ of a network, which can be interpreted as the minimum number of units to define as many linear regions, thereby providing a more practical and interpretable metric for expressiveness. We present the following contributions:

(i) We adapt approximate model counting methods for propositional satisfiability (SAT) to obtain probabilistic bounds on the number of solutions of MILP formulations, which we use to count regions. Interestingly, these methods are particularly simpler and faster when restricted to lower bounds on the order of magnitude. See results in Figure 2 and algorithm in Section 5.

(ii) We refine the best known upper bound by considering the coefficients of the trained network. With such information, we identify that unit activity further contributes to the bottleneck effect caused by narrow layers (Serra et al., 2018). Furthermore, we are able to compare networks with the same configuration of layers. See results in Table 1 and theory in Section 4.

(iii) We also survey and contribute to the literature on MILP formulations of rectifier networks due to the impact of the formulation on obtaining better empirical bounds. See Section 3.

## 2 PRELIMINARIES AND NOTATIONS

In this paper, we consider feedforward Deep Neural Networks (DNNs) with Rectifier Linear Unit (ReLU) activations. Each network has $n_0$ input variables given by $\boldsymbol{x} = [x_1 \ x_2 \ \ldots \ x_{n_0}]^T$ with a bounded domain $\mathbb{X}$ and $m$ output variables given by $\boldsymbol{y} = [y_1 \ y_2 \ \ldots \ y_m]^T$. Each hidden layer $l = \{1, 2, \ldots, L\}$ has $n_l$ hidden neurons with outputs given by $\boldsymbol{h}^l = [h_1^l \ h_2^l \ldots h_{n_l}^l]^T$. For notation simplicity, we may use $\boldsymbol{h}^0$ for $\boldsymbol{x}$ and $\boldsymbol{h}^{L+1}$ for $\boldsymbol{y}$. Let $\boldsymbol{W}^l$ be the $n_l \times n_{l-1}$ matrix where each row corresponds to the weights of a neuron of layer $l$. Let $\boldsymbol{b}^l$ be the bias vector used to obtain the activation functions of neurons in layer $l$. The output of unit $i$ in layer $l$ consists of an affine transformation $g_i^l = \boldsymbol{W}_i^l \boldsymbol{h}^{l-1} + \boldsymbol{b}_i^l$ to which we apply the ReLU activation $h_i^l = \max\{0, g_i^l\}$.

We may regard the DNN as a piecewise linear function $F : \mathbb{R}^{n_0} \to \mathbb{R}^m$ that maps the input $\boldsymbol{x} \in \mathbb{X} \subset \mathbb{R}^{n_0}$ to $\boldsymbol{y} \in \mathbb{R}^m$. Hence, the domain is partitioned into regions within which $F$ corresponds to an affine function, which we denote as linear regions. Following the same convention as Raghu et al. (2017); Montúfar (2017); Serra et al. (2018), we characterize each linear region by the set of units that are active in that domain. For each layer $l$, let $\mathbb{S}^l \subseteq \{1, \ldots, n_l\}$ be the activation set in which $i \in S^l$ if and only if $h_i^l > 0$. Let $\mathcal{S} = (\mathbb{S}^1, \ldots, \mathbb{S}^l)$ be the activation pattern aggregating those

activation sets. Consequently, the number of linear regions defined by the DNN corresponds to the number of nonempty sets in $\boldsymbol{x}$ defined by all possible activation patterns.

## 3 COUNTING AND MILP FORMULATIONS

We can represent each linear region defined by a rectifier network with $n$ hidden units on domain $\mathbb{X}$ by a distinct vector in $\{0, 1\}^n$, where each element denotes if the corresponding unit is active or not. Serra et al. (2018) have shown that such vector can be embedded into an MILP formulation mapping inputs to outputs of a rectifier network. For a neuron $i$ in layer $l$, this mapping uses such binary variable $z_i$, the vector $\boldsymbol{h}^{l-1}$ of inputs coming from layer $l - 1$, the variable $g_i^l$ for the value of the affine transformation $\boldsymbol{W}_i^l \boldsymbol{h}^{l-1} + \boldsymbol{b}_i^l$, the variable $h_i^l = \max\{0, g_i^l\}$ denoting the output of the unit, and a variable $\bar{h}_i^l$ denoting the output of a complementary fictitious unit $\bar{h}_i^l = max\{0, -g_i^l\}$:

$$\boldsymbol{W}_i^l \boldsymbol{h}^{l-1} + b_i^l = g_i^l \tag{1}$$

$$g_i^l = h_i^l - \bar{h}_i^l \tag{2}$$

$$h_i^l \leq H_i^l z_i^l \tag{3}$$

$$\bar{h}_i^l \leq \bar{H}_i^l (1 - z_i^l) \tag{4}$$

$$h_i^l \geq 0 \tag{5}$$

$$\bar{h}_i^l \geq 0 \tag{6}$$

$$z_i^l \in \{0, 1\} \tag{7}$$

For correctness, constants $H_i^l$ and $\bar{H}_i^l$ should be as large as $h_i^l$ and $\bar{h}_i^l$ can be. In such case, the value of $g_i^l$ determines if the unit or its fictitious complement is active. Note, however, that constraints (1)–(7) allow $z_i^l = 1$ when $g_i^l = 0$. To count the number of linear regions, Serra et al. (2018) uses the projection on the binary variables of the solutions where all active units have positive outputs, i.e., $h_i^l > 0$ if $z_i^l = 1$, thereby counting the positive solutions with respect to $f$ on the binary variables of

$$\max f \tag{8}$$

$$\text{s.t. } (1) - (7) \qquad l = 1, \ldots, L; \ i = 1, \ldots, n_l \tag{9}$$

$$f \leq h_i^l + (1 - z_i^l) H_i^l \qquad l = 1, \ldots, L; \ i = 1, \ldots, n_l : H_i^l > 0 \tag{10}$$

$$\boldsymbol{x} \in \mathbb{X} \tag{11}$$

The solutions of this projection can be enumerated using the one-tree algorithm (Danna et al., 2007), in which the branch-and-bound tree used to obtain the optimal solution is further expanded to collect near-optimal solutions up to a given limit. In general, finding a feasible solution to a MILP is NP-complete (Cook, 1971) and thus optimization is NP-hard. However, Fischetti & Jo (2018) note that a feasible solution can always be obtained by evaluating any valid input. While that does not directly imply that optimization problems on DNNs are easy, it hints at the possibility of good properties.

Several MILP formulations with an equivalent feasible set have been used in the context of network verification to determine the image of the function modeled (Lomuscio & Maganti, 2017; Dutta et al., 2018) and evaluate adversarial perturbations in the domain $\mathbb{X}$ (Cheng et al., 2017; Fischetti & Jo, 2018; Tjeng et al., 2017; Xiao et al., 2018). There are also similar applications relaxing the binary variables as continuous variables in the domain $[0, 1]$ or using the linear formulation of a particular linear region (Bastani et al., 2016; Ehlers, 2017; Wong & Kolter, 2018), which can be simply defined using $\boldsymbol{W}_i^l \boldsymbol{h}^{l-1} + b_i^l \geq 0$ for active units and the complement for inactive units.

Although equivalent, some authors have explored how these formulations may differ in strength (Fischetti & Jo, 2018; Tjeng et al., 2017; Huchette, 2018). When the binary variables are relaxed as continuous variables in the domain $[0, 1]$, we obtain a linear relaxation that may differ across formulations. We say that an MILP formulation $A$ is stronger than another formulation $B$ if, when projected on a common set of variables, the linear relaxation of $A$ is a proper subset of the linear relaxation of $B$. Formulation strength is commonly regarded as a proxy for MILP solver performance.

Differences in strength may be due to changes in constants such as $H_i^l$ and $\bar{H}_i^l$, use of additional valid inequalities that remove fractional solutions, or even additional variables defining an extended formulation. For mapping DNNs, we can discuss strength in at least three levels of scope.

First, we can consider the strength of the formulation to represent a ReLU activation $h_i^l = \max\{0, g_i^l\}$. Ideally, we want the projection on $g_i^l$ and $h_i^l$ to be the convex outer approximation of all possible combined values of those variables (Wong & Kolter, 2018), as illustrated in Figure 1 (a) and (b), which is in fact the case if the values of $H_i^l$ and $\bar{H}_i^l$ are the smallest possible.

**Lemma 1.** *If $H_i^l = \arg\max_{g^{l-1}}\{g_i^l\} \geq 0$ and $\bar{H}_i^l = \arg\max_{g^{l-1}}\{-g_i^l\} \geq 0$, then the linear relaxation of (2)–(7) defines the convex outer approximation on $(g_i^l, h_i^l)$.*

Lemma 1 evidences that, for formulations like the one above, constants for both the maximum and the minimum values of $h_i^l$ are necessary to obtain a strong formulation. The proof can be found in Appendix A. We note that a similar claim without proof is made by Huchette (2018).

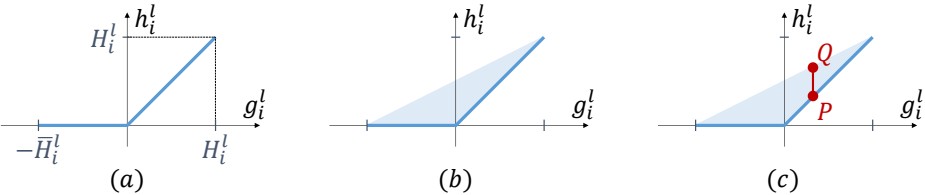

Figure 1: (a) ReLU mapping $h_i^l = \max\{0, g_i^l\}$; (b) Convex outer approximation on $(g_i^l, h_i^l)$; and (c) If P maps a vertex $x_P$ of the input, by convexifying the layer the rest of PQ is infeasible for $x_P$.

When the domain $\mathbb{X}$ of the network is defined by a box, in which case the domain of each input variable $x_i$ is an independent continuous interval, then the smallest possible values for $H_i^1$ and $\bar{H}_i^1$ can be computed with interval arithmetic by taking element-wise maxima (Cheng et al., 2017; Serra et al., 2018). When extended to subsequent layers, however, this approach is prone to overestimate the values for $H_i^l$ and $\bar{H}_i^l$ because the output of subsequent layers is not necessarily a box in which the maximum value of each input are independent from each other. More generally, if $\mathbb{X}$ is polyhedral, Fischetti & Jo (2018) and Tjeng et al. (2017) show that we can obtain the smallest values for these constants by solving a sequence of MILPs on layers $l' = 1, \dots, L$ of the form

$$H_i^{l'} = \max \qquad g_i^{l'} \tag{12}$$
$$\text{s.t.} \qquad (1) - (7) \qquad l = 1, \dots, l'-1; \; i = 1, \dots, n_l \tag{13}$$
$$x \in X \tag{14}$$

and replacing (equation 12) with $\max -g_i^{l'}$ to compute $\bar{H}_i^{l'}$. In large rectifier networks, Tjeng et al. (2017) found that many units are always active because $\bar{H}_i^l < 0$ or always inactive because $H_i^l \leq 0$. At the very least, we can use 0 on either case. In the former case, which they denote as *stably active*, we can simply replace constraints (1)-(7) with $h_i^l = g_i^l$. In the latter case, which they denote as *stably inactive*, we note that the unit can be removed from the formulation without any loss. They denote as *unstable* the remaining units, which can be active or not depending on their inputs.

Second, we can consider the strength of the formulation to represent the mapping of $\boldsymbol{h}^{l-1}$ to $\boldsymbol{h}^l$ on each layer. Huchette (2018) argues that this additional strengthening may remove certain combinations of $\boldsymbol{h}^{l-1}$ and $h_i^l$ that can never occur, and has shown that this can be done using an extended formulation following Balas (1998). Figure 1 (c) describes one such example. However, we observed a slower performance to count linear regions due to the larger number of variables. Huchette (2018) has also shown that these variables can be projected out, with the resulting formulation having an exponential number of constraints on $n_l$. In the context of finding a single optimal solution, usually not requiring all of them, these constraints can be efficiently generated as needed.

Third, we can consider constraints strengthening the formulation across different layers. For example, Huchette (2018) presents such a family of valid inequalities that resemble those obtained by projecting out the extra variables after convexifying each layer as described above.

### 3.1 BOUNDING OUTPUTS WITH ACTIVATIONS FROM THE PREVIOUS LAYER

We propose some valid inequalities involving consecutive layers of the network. The first is inspired by how constants $H_i^l$ and $\bar{H}_i^l$ can be bounded using interval arithmetic. Depending on which units

are active in the previous layer, the output of a given unit may be further restricted as follows:

$$h_i^l \leq \max\left\{0, b_i^l\right\} + \sum_{j \in \{1,\dots,n_{l-1}\}: \boldsymbol{W}_{ij}^l > 0} \boldsymbol{W}_{ij}^l H_j^{l-1} z_j^{l-1} \qquad l = 2, \dots, L; i = 1, \dots, n_l \qquad (15)$$

The $\max$ term is necessary in case $b_i^l$ is negative, since none of the units on the summation term being negative merely implies that the unit itself is inactive instead of rendering the system infeasible.

Following the same logic, we may actually define inequalities on the binary variables alone, which may be preferable since large constants create numerical difficulties and deteriorate solver performance. For the unit to be active when $b_i^l \leq 0$, there must be a positive contribution from the previous layer, and thus some unit $j$ in layer $l - 1$ such that $\boldsymbol{W}_{ij}^l > 0$ should be also active:

$$z_i^l \leq \sum_{j \in \{1,\dots,n_{l-1}\}: \boldsymbol{W}_{ij}^l > 0} z_j^{l-1} \qquad l = 2, \dots, L; i = 1, \dots, n_l : b_i^l \leq 0 \qquad (16)$$

Similarly, unit $i$ is only inactive when $b_i^l > 0$ if some unit $j$ in layer $l - 1$ such that $\boldsymbol{W}_{ij}^l < 0$ is active:

$$(1 - z_i^l) \leq \sum_{j \in \{1,\dots,n_{l-1}\}: \boldsymbol{W}_{ij}^l < 0} z_j^{l-1} \qquad l = 2, \dots, L; i = 1, \dots, n_l : b_i^l > 0 \qquad (17)$$

Let us denote unstable units in which $b_i^l \leq 0$, and thus (16) applies, as *inactive leaning*; and those in which $b_i^l > 0$, and thus (17) applies, as *active leaning*. Within linear regions where none among the units of the previous layer in the corresponding inequalities is active, these units can be regarded as stably inactive and stably active, respectively. We will use that to obtain better bounds in Section 4.

# 4 UPPER BOUND ON A PARTICULAR NETWORK

We prove a tighter – and empirical – upper bound in this section. This bound is obtained by taking into accounnt which units are stably active and stably inactive on the input domain $\mathbb{X}$ and also how many among the unstable units are locally stable in some of the linear regions. Prior to discussing this bound in Section 4.2 and how to compute its parameters in Section 4.3, we discuss in Section 4.1 other factors that have been found to affect such bounds in prior work on this topic.

## 4.1 FACTORS AFFECTING THE BOUND ON A CONFIGURATION OF LAYERS

The two main building blocks to bound the number of linear regions are activation hyperplanes and the theory of hyperplane arrangements. We explain their use in prior work in this Section.

For each unit $i$ in layer $l$, the activation hyperplane $\boldsymbol{W}_i^l \boldsymbol{h}^{l-1} + b_i^l = 0$ splits the input space $\boldsymbol{h}^{l-1}$ into the regions where the unit is active ($\boldsymbol{W}_i^l \boldsymbol{h}^{l-1} + b_i^l > 0$) or inactive ($\boldsymbol{W}_i^l \boldsymbol{h}^{l-1} + b_i^l \leq 0$). In order to bound the number of regions defined by multiple hyperplanes on the same space, we use a result from Zaslavsky (1975) that $n_l$ hyperplanes in an $n_{l-1}$-dimensional space define at most $\sum_{j=0}^{n_{l-1}} \binom{n_l}{j}$ regions. However, if the normal vectors of these hyperplanes span a smaller space, then the same number of regions can be defined in less dimensions. In particular, Serra et al. (2018) shows that we can actually assume a maximum of $\sum_{j=0}^{\text{rank}(\boldsymbol{W}^l)} \binom{n_l}{j} \leq \sum_{j=0}^{\min\{n_{l-1}, n_l\}} \binom{n_l}{j}$ regions instead.

We can obtain a bound for deep networks by recursively combining the bounds obtained on each layer. By assuming that every linear region defined by the first $l - 1$ layers is then subdivided into the maximum possible number of linear regions defined by the activation hyperplanes of layer $l$, we obtain the implicit bound of $\prod_{l=1}^{L} \sum_{j=0}^{n_{l-1}} \binom{n_l}{j}$ from Raghu et al. (2017). By observing that the dimension of the input of layer $l$ on each linear region is also constrained by the smallest input dimension among layers 1 to $l-1$, we can obtain the bound in Montúfar (2017) of $\prod_{l=1}^{L} \sum_{j=0}^{d_l} \binom{n_l}{j}$, where $d_l = \min\{n_0, n_1, \dots, n_l\}$. If we refine the effect on the input dimension by also considering that the number of units that are active on each layer varies across the linear regions, we can obtain the tighter bound in Serra et al. (2018) of $\sum_{(j_1,\dots,j_L) \in J} \prod_{l=1}^{L} \binom{n_l}{j_l}$, where $J = \{(j_1, \dots, j_L) \in \mathbb{Z}^L : 0 \leq j_l \leq \min\{n_0, n_1 - j_1, \dots, n_{l-1} - j_{l-1}, n_l\} \ \forall l = 1, \dots, L\}$.

## 4.2 FACTORS AFFECTING THE BOUND ON A PARTICULAR NETWORK

Now we show that we can further improve on the sequence of bounds previously found in the literature by leveraging the local and global stability of units of a trained network, which can be particularly useful to compare networks having the same configuration of layers.

First, note that only units that can be active in a given linear region produced by layers 1 to $l-1$ affect the dimension of the space in which the linear region can be further partitioned by layers $l$ to $L$. Second, only the subset of these units that can also be inactive within that region, i.e., the unstable ones, counts toward the number of hyperplanes partitioning the linear region at layer $l$. Hence, let $\mathcal{A}_l(k)$ be the maximum number of units that can be active in layer $l$ if $k$ units are active in layer $l-1$; and $\mathcal{I}_l(k)$ be the corresponding maximum number of units that are unstable, hence potentially defining hyperplanes that intersect the interior of the linear region. Note that every linear region is contained in one side of the hyperplane defined by each stable unit. We state our main result below and discuss how to compute $\mathcal{A}_l(k)$ and $\mathcal{I}_l(k)$ using $\boldsymbol{W}^l$ and $\boldsymbol{b}^l$ next.

Theorem 2 improves the result by Serra et al. (2018) when not all hyperplanes partition every linear region from previous layers ($\mathcal{I}_l(k_{l-1}) < n_l$) or not all units can be active (smaller intervals for $j_l$):

**Theorem 2.** *Consider a deep rectifier network with $L$ layers with input dimension $n_0$ and at most $\mathcal{A}_l(k)$ active units and $\mathcal{I}_l(k)$ unstable units in layer $l$ for every linear region defined by layers 1 to $l-1$ when $k$ units are active in layer $l-1$. Then the maximum number of linear regions is at most*

$$\sum_{(j_1,\ldots,j_L)\in J} \prod_{l=1}^{L} \binom{\mathcal{I}_l(k_{l-1})}{j_l}$$

*where $J = \{(j_1,\ldots,j_L) \in \mathbb{Z}^L : 0 \leq j_l \leq \min\{n_0, k_1,\ldots,k_{l-1}, \mathcal{I}_l(k_{l-1})\}\}$ with $k_0 = n_0$ and $k_l = \mathcal{A}_l(k_{l-1}) - j_{l-1}$ for $l = 1,\ldots,L$.*

*Proof.* In resemblance to Serra et al. (2018), we define a recurrence to recursively bound the number of subregions within a region. Let $R(l,k,d)$ be an upper bound to the maximal number of regions attainable from partitioning a region with dimension at most $d$ among those defined by layers 1 to $l-1$ in which at most $k$ units are active in layer $l-1$ by using the remaining layers $l$ to $L$. For the base case $l = L$, we have $R(L,k,d) = \sum_{j=0}^{\min\{\mathcal{I}_L(k),d\}} \binom{\mathcal{I}_L(k)}{j}$ since $\mathcal{I}_l(k) \leq \mathcal{A}_l(k)$. The recurrence groups regions with same number of active units in layer $l$ as $R(l,k,d) = \sum_{j=0}^{\mathcal{A}_l(k)} N^l_{\mathcal{I}_l(k),d,j} R(l+1, j, \min\{j,d\})$ for $l = 1$ to $L-1$, where $N^l_{p,d,j}$ represents the maximum number of regions with $j$ active units in layer $l$ from partitioning a space of dimension $d$ using $p$ hyperplanes.

We also use the observation in Serra et al. (2018) that there are at most $\binom{\mathcal{I}_l(k)}{j}$ regions defined by layer $l$ when $j$ unstable units are active and there are $k$ active units in layer $l-1$, which can be regarded as the subsets of $\mathcal{I}_l(k)$ units of size $j$. Since layer $l$ defines at most $\sum_{j=0}^{\min\{\mathcal{I}_l(k),d\}} \binom{\mathcal{I}_l(k)}{j}$ regions with an input dimension $d$ and $k$ active units above, by allowing the largest number of active hyperplanes among the unstable units and also using $\binom{\mathcal{I}_l(k)}{\mathcal{I}_l(k)-j} = \binom{\mathcal{I}_l(k)}{j}$, we have

$$R(l,k,d) = \begin{cases} \displaystyle\sum_{j=0}^{\min\{\mathcal{I}_l(k),d\}} \binom{\mathcal{I}_l(k)}{j} R(l+1, \mathcal{A}_l(k)-j, \min\{\mathcal{A}_l(k)-j, d\}) & \text{if } 1 \leq l \leq L-1, \\ \displaystyle\sum_{j=0}^{\min\{\mathcal{I}_L(k),d\}} \binom{\mathcal{I}_L(k)}{j} & \text{if } l = L. \end{cases}$$

Without loss of generality, we assume that the input is generated by $n_0$ active units feeding the network, hence implying that the bound can be evaluated as $R(1, n_0, n_0)$:

$$\sum_{j_1=0}^{\min\{\mathcal{I}_1(k_0),d_1\}} \binom{\mathcal{I}_1(k_0)}{j_1} \sum_{j_2=0}^{\min\{\mathcal{I}_2(k_1),d_2\}} \binom{\mathcal{I}_2(k_1)}{j_2} \cdots \sum_{j_L=0}^{\min\{\mathcal{I}_L(k_{L-1}),d_L\}} \binom{\mathcal{I}_L(k_{L-1})}{j_L}$$

where $k_0 = n_0$ and $k_l = \mathcal{A}_l(k_{l-1}) - j_{l-1}$ for $l = 1,\ldots,L$, whereas $d_l = \min\{n_0, k_1,\ldots,k_{l-1}\}$. We obtain the final expression by nesting the values of $j_1,\ldots,j_L$. □

## 4.3 BOUNDING ACTIVE AND UNSTABLE UNITS ACROSS ACTIVATION SETS

Finally, we discuss how the parameters introduced with the empirical bound in Section 4.2 can be computed exactly, or else approximated. We first bound the value of $\mathcal{I}_l(k)$. Let $U_l^-$ and $U_l^+$ denote the sets of inactive leaning and active leaning units in layer $l$, and $U_l = U_l^+ \cup U_l^-$. For a given unit $i \in U_l^-$, we can define a set $J^-(l, i)$ of units from layer $l - 1$ that, if active, can potentially make $i$ active. In fact, we can define the set in the summation of inequality (16), and therefore let $J^-(l, i) := \{j : 1 \leq j \leq n_{l-1}, \boldsymbol{W}_{ij}^l > 0\}$. For a given unit $i \in U_l^+$, we can similarly use the set in inequality (17), and let $J^+(l, i) := \{j : 1 \leq j \leq n_{l-1}, \boldsymbol{W}_{ij}^l < 0\}$. Conversely, let $I(l, j) := \{i : i \in U_{l+1}^+, j \in J^+(l+1, i)\} \cup \{i : i \in U_{l+1}^-, j \in J^-(l+1, i)\}$ be the set of units in layer $l + 1$ that may be locally unstable if unit $j$ in layer $l$ is active.

**Proposition 3.** $\mathcal{I}_l(k) \leq \max\limits_{S} \left\{ \left| \bigcup\limits_{j \in S} I(l-1, j) \right| : S \subseteq \{1, \ldots, n_{l-1}\}, |S| \leq k \right\}$

In other words, we look for the subsets of at most $k$ units in layer $l - 1$ that together may affect the stability of the largest number of units in layer $l$. Nonetheless, we may only need to inspect a small number of such subsets in practice. Assuming that each row of $\boldsymbol{W}^l$ and vector $\boldsymbol{b}^l$ have about the same number of positive and negative elements, then we can expect that each set $I(l-1, j)$ contains half of the units in $U_l$. If these positive and negative elements are distributed randomly for each unit, then a logarithmic number of the units in layer $l - 1$ being active may suffice to entirely cover $U_l$. Hence, we can reasonably expect to evaluate a linear number of subsets of $n_{l-1}$ on average. In cases where this assumption does not hold, we discuss later how to approximate $\mathcal{I}_l(k)$.

Next we bound the value of $\mathcal{A}_l(k)$. In this case, we consider a larger subset of the units in $l$ that only excludes locally inactive units. Let $n_l^+$ denote the number of stably active units in layer $l$, which is such that $n_l^+ \leq n_l - |U_l|$, and let $I^-(l, j) := \{i : i \in U_{l+1}^-, j \in J^-(l+1, i)\}$ be the set of inactive leaning units in layer $l + 1$ that may become active when unit $j$ in layer $l$ is active.

**Proposition 4.** $\mathcal{A}_l(k) \leq n_l^+ + |U_l^+| + \max\limits_{S} \left\{ \left| \bigcup\limits_{j \in S} I^-(l-1, j) \right| : S \subseteq \{1, \ldots, n_{l-1}\}, |S| \leq k \right\}$

We can approximate $\mathcal{I}_l(k)$ and $\mathcal{A}_l(k)$ with strong optimality guarantees $(1 - \frac{1}{e})$ using simple greedy algorithms for submodular function maximization (Nemhauser et al., 1978). See Appendix E.

## 5 APPROXIMATE MODEL COUNTING: FROM SAT TO MILP

We can think of SAT as a particular form of encoding solutions on a set $V$ of Boolean variables, where the solutions have to satisfy a set of predicates, and which can therefore represent solutions on binary variables of an MILP. Toda (1989) has shown that counting solutions of SAT formulas is #P-complete. However, thanks to the improving performance of SAT solvers, many practical approaches to approximate the number of solutions have been proposed since Gomes et al. (2006a), all of which making a relatively small number of solver calls to solve restricted formulas.

The idea in this line of work is to use hash functions with good statistical properties to partition the set of solutions $S$ into subsets having approximately half of the solutions each. After restricting a given formula to one of such subsets $r$ times, we may intuitively assume that, with some probability, $|S| \geq 2^r$ if the resulting subset is more often feasible or else $|S| < 2^r$. Most of the literature has restricted SAT formulas with predicates that encode XOR constraints, which can be interpreted in terms of 0–1 variables as restricting the sum of a subset $U$ of the variables to be even or odd. Probabilistic lower bounds can be obtained using XOR constraints on fixed or variable sizes of subset $k = |U|$. Although they get better as $k$ increases, even small values of $k$ yield good approximations in practice (Gomes et al., 2007b). Since we are mainly interested in the order of magnitude, we focus on extending the classic MBound algorithm (Gomes et al., 2006a). We opt for a fixed – and also small – size $k$ to avoid scalability issues as the number of ReLUs increase. We refer the reader to Appendix C for a survey on XOR constraints and approximate model counting.

The key difference when devising an algorithm for MILP is that these solvers are not used in the same way as SAT solvers. The assumption in SAT-based approaches is that each restricted formula

entails a new call to the solver. Chakraborty et al. (2016) improves to a logarithmic number of calls by orderly applying the same sequence constraints up to each value of $r$, and then applying binary search to find the smallest $r$ that makes the formula unsatisfiable. In MILP solvers, we can test for all values of $r$ with a single call to the solver by generating parity constraints as lazy cuts, which can be implemented through callbacks. When a new solution is found, a callback is invoked to generate parity constraints. Each constraint may or may not remove the solution just found, since we preserve the independence between the solutions found and the constraints generated, and thus we may need to generate multiple parity constraints before yielding the control back to the solver. Algorithm 1, which we denote MIPBound, illustrates the idea. We refer the reader to Appendix C for details on how to translate parity constraints to MILP and Appendix D for how the probabilities are derived.

## 6 EXPERIMENTS

We test on the instances used in Serra et al. (2018) to benchmark against exact counting. The results are reported in Figure 2 and Table 1. We adapt Algorithm 1 to count linear regions by ignoring solutions with value 0. For each size of parity constraints $k$, which we denote as XOR-$k$, we measure the time to find the smallest coefficients $H_i^l$ and $\bar{H}_i^l$ for each unit along with the subsequent time of Algorithm 1. We let Algorithm 1 run for enough steps to obtain a probability of 99.5% in case all tested restrictions of a given size preserve the formulation feasible, and we report the largest lower bound with probability at least 95%. We define a DNN with $\eta < 12$ as small and large otherwise to illustrate how these points are distributed with respect to the identity line, since counting is faster than sampling for smaller sets. The upper bound from Theorem 2, which we denote as Empirical Upper Bound (Empirical UB), is computed at a fraction of the time to obtain the constants. We use Configuration Upper Bound (Configuration UB) for the bound in Serra et al. (2018). The code is written in C++ (gcc 4.8.4) using CPLEX Studio 12.8 as a solver and ran in Ubuntu 14.04.4 on a machine with 40 Intel(R) Xeon(R) CPU E5-2640 v4 @ 2.40GHz processors and 132 GB of RAM.

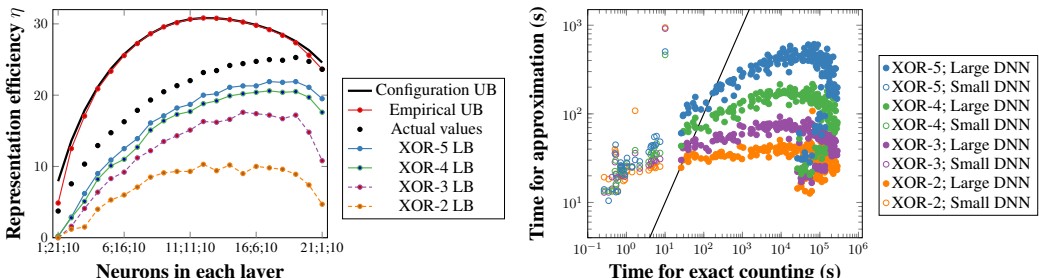

Figure 2: Left: averages of the proposed Empirical UB and XOR-$k$ lower bounds with probability 95% compared with the Configuration UB and the actual number of regions reported in Serra et al. (2018) for 10 networks of each type. Right: comparison of approximations vs. exact counting times.

| 1;21;10 | 2;20;10 | 3;19;10 | | | | | | . . . | | | | | | | 19;3;10 | 20;2;10 | 21;1;10 |
|---|---|---|---|---|---|---|---|---|---|---|---|---|---|---|---|---|---|
| 73.1 | 17.8 | 10.4 | 3.1 | 3.9 | 2 | 1.1 | 0 | 0 | 1 | 0 | 0 | 0.1 | 0.2 | 0.5 | 1 | 1.8 | 3.4 | 9.5 | 44.5 | 98.3 |

Table 1: Gap between configuration UB and actual values that is closed (%) by empirical UB.

Following up on the discussion from Section 4.3 about computing the values of $\mathcal{A}_l(k)$ and $\mathcal{I}_l(k)$, we report in Table 2 of Appendix F the minimum value of $k$ to find the maximum value of both expressions for layers 2 and 3 of the trained networks. We observe in practice that such values of $k$ remain small and so does the number of subsets of units from layer $l-1$ that we need to inspect.

## 7 CONCLUSION

This paper introduced methods to obtain upper and lower bounds on a rectifier network. The upper bound refines the best known result for the network configuration by taking into account the coefficients of the network. By analyzing how the network coefficients affect when each unit can be

active, we break the commonly used theoretical assumption that the activation hyperplane of each unit intersects every linear region defined by the previous layers. The resulting bound is particularly stronger when the network has a narrow layer, hence evidencing that the bottleneck effected identified by Serra et al. (2018) can be even stronger in those cases. The lower bound is based on extending an approximate model counting algorithm of SAT formulas to MILP formulations, which can then be used on MILP formulations of rectifier networks. The resulting algorithm is orders of magnitude faster than exact counting on networks with a large number of linear regions. The probabilistic bounds obtained can be parameterized for a balance between precision and speed, but it is interesting to observe that the the bounds obtained for different networks preserve a certain ordering in their sizes as we make the estimate more precise. Hence, we have some indication that faster approximations could suffice if we just want to compare networks for their relative expressiveness.

---

**Algorithm 1** Computes probabilistic lower bounds on the number of distinct solutions on $n$ binary variables of a formulation $F$ using parity constraints of size $k$

---
1: **function** MIPBOUND($F, n, k$)
2:     $i \leftarrow 0$
3:     **for** $j \leftarrow 0 \rightarrow n$ **do**
4:         $f[j] \leftarrow 0$
5:     **end for**
6:     **while** Termination criterion not satisfied **do**
7:         $F' \leftarrow F$                                                     ▷ Start over with $F'$ as formulation $F$
8:         $i \leftarrow i + 1$                                           ▷ Number of times that we have made $F'$ infeasible
9:         $r \leftarrow 0$                                                   ▷ Number of parity constraints added this time
10:         **while** $F'$ has some solution $s$ **do**
11:             **repeat**
12:                 Generate parity constraint $C$ of size $k$ among $n$ variables
13:                 $F' \leftarrow F' \cap C$
14:                 $r \leftarrow r + 1$
15:             **until** $C$ removes $s$                             ▷ This loop is implemented as a lazy cut callback
16:         **end while**
17:         **for** $j \leftarrow 0 \rightarrow r - 1$ **do**
18:             $f[j] \leftarrow f[j] + 1$                   ▷ Number of times that $F'$ is feasible after adding $j$ constraints
19:         **end for**
20:     **end while**
21:     **for** $j \leftarrow 0 \rightarrow n - 1$ **do**                                 ▷ Computes probabilities after last call to the solver
22:         $\delta \leftarrow f[j+1]/i - 1/2$
23:         **if** $\delta > 0$ **then**                                 ▷ Formulation after $j + 1$ constraints is more often feasible
24:             $P_j \leftarrow 1 - \left( \frac{e^{2.\delta}}{(1+2.\delta)^{1+2.\delta}} \right)^{i/2}$                      ▷ Probability that $|S| > 2^j$
25:         **else**                                 ▷ If formulation is more often infeasible, then no probability is defined
26:             **break**                                           ▷ Same is true for subsequent values of $j$, so exit loop
27:         **end if**
28:     **end for**
29:     **return** Probabilities $P$
30: **end function**

---

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

## A  CONVEX OUTER APPROXIMATION OF A BOUNDED UNIT

**Lemma 1.** *If $H_i^l = \arg\max_{h^{l-1}}\{W_i^l h^{l-1} + b_i^l\} \geq 0$ and $\bar{H}_i^l = \arg\max_{h^{l-1}}\{-W_i^l h^{l-1} - b_i^l\} \geq 0$, then the linear relaxation of (1)–(7) defines the convex outer approximation on $(g_i^l, h_i^l)$.*

*Proof.* We begin with the linear relaxation of the formulation defined by constraints (2)–(7):

$$g_i^l = h_i^l - \bar{h}_i^l \qquad \Leftrightarrow \qquad \bar{h}_i^l = h_i^l - g_i^l \tag{18}$$

$$h_i^l \leq H_i^l z_i^l \qquad \Leftrightarrow \qquad z_i^l \geq \frac{h_i^l}{H_i^l} \tag{19}$$

$$\bar{h}_i^l \leq \bar{H}_i^l (1 - z_i^l) \qquad \Leftrightarrow \qquad z_i^l \leq 1 - \frac{\bar{h}_i^l}{\bar{H}_i^l} \tag{20}$$

$$h_i^l \geq 0 \tag{21}$$

$$\bar{h}_i^l \geq 0 \tag{22}$$

$$0 \leq z_i^l \leq 1 \tag{23}$$

We first project $z_i^l$ out by isolating that variable on one side of each inequality, and then combining every lower bound with every upper bound. Hence, we replace (19), (20), and (23) with:

$$\frac{h_i^l}{H_i^l} \leq 1 - \frac{\bar{h}_i^l}{\bar{H}_i^l} \qquad \Leftrightarrow \qquad \bar{h}_i^l \leq \bar{H}_i^l \left(1 - \frac{h_i^l}{H_i^l}\right) \tag{24}$$

$$\frac{h_i^l}{H_i^l} \leq 1 \qquad \Leftrightarrow \qquad h_i^l \leq H_i^l \tag{25}$$

$$0 \leq 1 - \frac{\bar{h}_i^l}{\bar{H}_i^l} \qquad \Leftrightarrow \qquad \bar{h}_i^l \leq \bar{H}_i^l \tag{26}$$

$$0 \leq 1 \tag{27}$$

Next, we project $\bar{h}_i^l$ through the same steps, also combining the equality with the lower and upper bounds on the variable. Hence, we replace (18), (22), (24), and (26) with:

$$h_i^l - g_i^l \geq 0 \tag{28}$$

$$h_i^l - g_i^l \leq \bar{H}_i^l \left(1 - \frac{h_i^l}{H_i^l}\right) \tag{29}$$

$$h_i^l - g_i^l \leq \bar{H}_i^l \tag{30}$$

$$\bar{H}_i^l \left(1 - \frac{h_i^l}{H_i^l}\right) \geq 0 \tag{31}$$

$$\bar{H}_i^l \geq 0 \tag{32}$$

We drop (27) as a tautology and (32) as implicit on our assumptions. Similarly, for $\bar{H}_i^l > 0$, inequality (31) is equivalent to (25). Therefore, we are left with (21), (25), (28), (29), and (30). We show in Figure 3 that the first four inequalities define the convex outer approximation on $(g_i^l, h_i^l)$, whereas (30) is active at $(-\bar{H}_i^l, 0)$ and $(H_i^l, H_i^l + \bar{H}_i^l)$ and thus dominated by (29) in that region.   $\square$

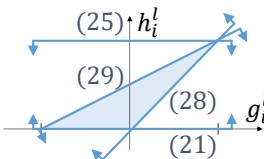

Figure 3: Using the same representation as in Figure 1, we illustrate how the projected inequalities (21), (25), (28), and (29) define the convex outer approximation on $(g_i^l, h_i^l)$.

## B  XOR CONSTRAINTS AND APPROXIMATE MODEL COUNTING

Carter & Wegman (1979) have shown that XOR constraints are universal hash functions. Furthermore, Sipser (1983) and Stockmeyer (1985) used these functions to show that approximate counting can be done in polynomial time with an NP-oracle, whereas Valiant & Vazirani (1986) have shown that SAT formulas with unique solution are as hard as those with multiple solutions. Hence, from a theoretical standpoint, such approximations are not much harder than solving for a single solution.

The seminal work by Gomes et al. (2006a) introduced the MBound algorithm, where XOR constraints on sets of variables with a fixed size $k$ are used to compute the probability that $2^r$ is either a lower or an upper bound. These probabilistic lower bounds are always valid but get better as $k$ increases, whereas the probabilistic upper bound is only valid if $k = |V|/2$. However, Gomes et al. (2007b) have shown that these lower bounds can be very good in practice for small values of $k$. The same principles have also been applied to constraint satisfaction problems (Gomes et al., 2007a).

With time, this topic has gradually shifted to more precise estimates and to reducing the value of $k$ needed to obtain valid upper bounds. Some of the subsequent work has been influenced by uniform sampling results from Gomes et al. (2006b), where the fixed size $k$ is replaced with an independent probability $p$ of including each variable in each XOR constraint. That work includes the ApproxMC and the WISH algorithms (Chakraborty et al., 2013; Ermon et al., 2013b), which rely on finding more solutions of the restricted formulas but generate $(\sigma, \epsilon)$ certificates by which, with probability $1 - \sigma$, the result is within $(1 \pm \epsilon)|S|$. The following work by Ermon et al. (2014) and Zhao et al. (2016) aimed at providing upper bound guarantees when $p < 1/2$, showing that the size of those sets can be $\Theta\big(log(|V|)\big)$. Other groups tackled this issue differently. Chakraborty et al. (2014) and Ivrii et al. (2016) have limited the counting to any set of variables $I$ for which any assignment leads to at most one solution in $V$, denoting those as minimal independent supports. Achlioptas & Jiang (2015) and Achlioptas et al. (2018) have broken with the independent probability $p$ by using each variable the same number of times across the $r$ XOR constraints.

## C  PARITY CONSTRAINTS IN MILP

Similarly to the case of SAT formulas, we need to find a suitable way of translating a XOR constraint to a MILP formulation. Let $w$ be the set of binary variables and $U \subseteq V$ the set of indices of $w$ variables of a XOR constraint. To remove all assignments to that subset of variables with an even sum, we can use a family of canonical cuts on the unit hypercube (Balas & Jeroslow, 1972):

$$\sum_{i \in U'} w_i - \sum_{i \in U \setminus U'} w_i \leq |U'| - 1 \qquad \forall U' \subseteq U : |U'| \text{ is even}, \tag{33}$$

which is effectively separating each such assignment with one constraint. Although exponential in $k$, Jeroslow (1975) has shown that each of those constraints – and only those – are necessary to define a convex hull of the feasible assignments in the absence of other constraints. However, we note that we can do better when $k = 2$ by using

$$w_i + w_j = 1 \qquad \text{if } U = \{\, i, j \,\}. \tag{34}$$

Due to the multiple XOR constraints used and the small $k$, we avoid moving away from the original space of variables. Alternatively, Yannakakis (1991) provides an extended formulation requiring a polynomial number of constraints. We note that these two possibilities have also been discussed by Ermon et al. (2013a) for a related application of probabilistic inference.

## D  DERIVING THE LOWER BOUND PROBABILITIES OF ALGORITHM 1

The probabilities given to the lower bounds by Algorithm 1 are due to the main result in Gomes et al. (2006a), which is based on the following parameters: XOR size $k$; number of restrictions $r$; loop repetitions $i$; number of repetitions that remain feasible after $j$ restrictions $f[j]$; deviation $\delta \in (0, 1/2]$; and precision slack $\alpha \geq 1$. We choose the values for the latter two.

A strict lower bound of $2^{r-\alpha}$ can be defined if

$$f[j] \geq i.(1/2 + \delta), \tag{35}$$

and for $\delta \in (0, 1/2)$ it holds with probability $1 - \left(\frac{e^\beta}{(1+\beta)^{1+\beta}}\right)^{i/2^\alpha}$ for $\beta = 2^\alpha.(1/2 + \delta) - 1$. We choose $\alpha = 1$, hence making $\beta = 2.\delta$, and then set $\delta$ to the largest value satisfying condition (35).

## E    APPROXIMATION ALGORITHMS FOR COMPUTING $A_l(k)$ AND $I_l(k)$

In section 4.3, we show the bounds for $\mathcal{I}_l(k)$ and $\mathcal{A}_l(k)$ as given below:

$$\mathcal{I}_l(k) \leq \max_S \left\{ \left| \bigcup_{j \in S} I(l-1, j) \right| : S \subseteq \{1, \ldots, n_{l-1}\}, |S| \leq k \right\}$$

$$\mathcal{A}_l(k) \leq n_l^+ + |U_l^+| + \max_S \left\{ \left| \bigcup_{j \in S} I^-(l-1, j) \right| : S \subseteq \{1, \ldots, n_{l-1}\}, |S| \leq k \right\}$$

The maximization terms on the right hand side of the inequalities for $\mathcal{I}_l(k)$ and $\mathcal{A}_l(k)$ can be seen as finding a set of $k$ subsets of the form $I(l-1, j)$ or $I^-(l-1, j)$, respectively, and whose union achieves the largest cardinality. This can be shown to be directly related to the maximum k-coverage problem with $(1 - \frac{1}{e})$−approximation using an efficient greedy algorithm (Feige, 1998). Note that the maximum k-coverage problem is actually a special case of the maximization of submodular functions, which are discrete analogue of convex functions (Nemhauser et al., 1978). For large networks, the use of greedy algorithms can be beneficial to get good approximations for $\mathcal{I}_l(k)$ and $\mathcal{A}_l(k)$ efficiently.

## F    THE VALUES OF $A_l(k)$ AND $I_l(k)$ IN TRAINED NETWORKS

| | Layer 2 | | | | | | Layer 3 | | | | | |
|---|---|---|---|---|---|---|---|---|---|---|---|---|
| | $\min \arg \max_k A_2(k)$ | | | $\min \arg \max_k I_2(k)$ | | | $\min \arg \max_k A_3(k)$ | | | $\min \arg \max_k I_3(k)$ | | |
| Configuration | Avg | Min | Max | Avg | Min | Max | Avg | Min | Max | Avg | Min | Max |
| 01;21;10 | 1.0 | 1 | 1 | 1.0 | 1 | 1 | 1.9 | 1 | 2 | 1.9 | 1 | 2 |
| 02;20;10 | 1.9 | 1 | 2 | 2.0 | 2 | 2 | 2.0 | 2 | 2 | 2.0 | 2 | 2 |
| 03;19;10 | 2.0 | 1 | 3 | 2.3 | 2 | 3 | 2.1 | 2 | 3 | 2.1 | 2 | 3 |
| 04;18;10 | 2.2 | 2 | 3 | 2.3 | 2 | 3 | 2.3 | 2 | 3 | 2.3 | 2 | 3 |
| 05;17;10 | 2.0 | 1 | 3 | 2.4 | 2 | 3 | 2.5 | 2 | 3 | 2.5 | 2 | 3 |
| 06;16;10 | 1.9 | 1 | 3 | 2.3 | 2 | 3 | 2.5 | 2 | 3 | 2.5 | 2 | 3 |
| 07;15;10 | 2.0 | 1 | 3 | 2.3 | 2 | 3 | 2.9 | 2 | 3 | 2.9 | 2 | 3 |
| 08;14;10 | 1.5 | 1 | 2 | 2.3 | 2 | 3 | 2.7 | 2 | 3 | 2.7 | 2 | 3 |
| 09;13;10 | 1.6 | 1 | 2 | 2.0 | 2 | 2 | 2.8 | 2 | 3 | 2.8 | 2 | 3 |
| 10;12;10 | 1.1 | 1 | 2 | 2.1 | 2 | 3 | 3.0 | 2 | 4 | 3.0 | 2 | 4 |
| 11;11;10 | 1.6 | 1 | 2 | 2.1 | 2 | 3 | 3.1 | 2 | 5 | 3.1 | 2 | 5 |
| 12;10;10 | 1.3 | 1 | 2 | 2.0 | 2 | 2 | 3.4 | 3 | 4 | 3.4 | 3 | 4 |
| 13;09;10 | 1.0 | 1 | 1 | 2.1 | 2 | 3 | 3.0 | 2 | 4 | 3.0 | 2 | 4 |
| 14;08;10 | 1.1 | 1 | 2 | 2.1 | 2 | 3 | 3.2 | 2 | 4 | 3.2 | 2 | 4 |
| 15;07;10 | 1.0 | 1 | 1 | 2.4 | 2 | 3 | 3.1 | 2 | 4 | 3.1 | 2 | 4 |
| 16;06;10 | 1.0 | 1 | 1 | 2.3 | 2 | 3 | 3.3 | 3 | 4 | 3.3 | 3 | 4 |
| 17;05;10 | 1.0 | 1 | 1 | 2.2 | 2 | 3 | 3.0 | 2 | 4 | 3.0 | 2 | 4 |
| 18;04;10 | 1.0 | 1 | 1 | 2.1 | 1 | 3 | 3.0 | 2 | 4 | 3.0 | 2 | 4 |
| 19;03;10 | 1.0 | 1 | 1 | 2.0 | 1 | 3 | 3.0 | 3 | 3 | 3.0 | 3 | 3 |
| 20;02;10 | 1.0 | 1 | 1 | 1.2 | 1 | 2 | 2.0 | 2 | 2 | 2.0 | 2 | 2 |
| 21;01;10 | 1.0 | 1 | 1 | 1.0 | 1 | 1 | 1.0 | 1 | 1 | 1.0 | 1 | 1 |

Table 2: Minimum value of $k$ to reach the maximum of $A_l(k)$ and $I_l(k)$ in the trained networks.

