# OpenReview forum: "Empirical Bounds on Linear Regions of Deep Rectifier Networks"
_ICLR.cc/2019/Conference_

### Official Review · AnonReviewer2 · 2018-10-31
**Interesting, but incremental and of little practical use**

**Rating:** 6
**Confidence:** 4

**Review:**

The paper deals with a problem of expressiveness of a piecewise linear neural network, characterized by the number of linear regions of the function modeled. This is one of the widely accepted measure of expressiveness of a linear model. As such, it has been studied before. The main contributions of the paper are:
1) Different algorithms are proposed that allow to compute the bounds faster, leveraging probabilistic algorithms
2) Tighter bounds are obtained
I find the results somewhat interesting. However, I do not think there is a lot of practical value in having faster algorithms for obtaining the bounds, as they are not used in practice anyway. I am also not convinced that the quest for tighter-and-tighter bounds in this approach is the right scientific direction. I find the paper to be an interesting contribution, but of a marginal value to the progress of the domain and for the improvement of our understanding of the models.

---

> ### Author Response · Authors · 2018-11-15
> **Answer to AnonReviewer2**
>
> Thank you for your time reviewing our paper. We appreciate your comments and we gave some thought on how to better address them. Please let us know if you have more questions or suggestions after reading our response below.
>
> You are correct in saying that the number of linear regions is not used in practice. We believe that, in part, because this number is expensive to compute. In the paper, we circumvent that by obtaining probabilistic lower bounds that get close to the actual number with much less effort.
>
> That is also the motivation for our upper bound, which exploits more information about the trained networks to compare networks with the same configuration of layers. We note that this is something that could not be done using the previous bounds. To do that, we are looking more closely at how the coefficients of the trained network determine the number of different activation patterns that a network may exhibit, hence helping us to better understand rectifier networks.
>
> To clarify that, we included the following line to contribution (ii) in the introduction:
>
> “Furthermore, we can compare networks with the same configuration of layers.”

---

### Official Review · AnonReviewer3 · 2018-11-02
**Well presented but highly technical paper on counting linear regions in neural networks**

**Rating:** 7
**Confidence:** 4

**Review:**

Summary:
This paper builds off of previous work that has studied the counting of linear regions in deep neural networks. The function learned by a deep neural network with piecewise linear activations (such as Relus) is itself piecewise linear on the input, and a measure of expressiveness of the network has been to count the number of linear regions.

However counting linear regions in a typical neural network is usually intractable, and there have been a sequence of upper and lower bounds proposed. Upper bounds are based on counting hyperplane arrangements (Zaslavsky, 1975; Raghu 2017; Montufar 2017; Serra 2018), and lower bounds based on counting regions in specific networks.

This paper improves the upper bound proposed in Serra (2018) by improving on a dimensionality constraint: the upper bound can be tightened if the dimensionality of the ambient space is shown to be smaller than the maximum possible value (number of neurons.) The paper defines A_l(k) -- the number of active neurons in layer l given k active neurons in layer l-1, and I_l(k) similar for inactive neurons, and proves an improved upper bound.

For the lower bound, the paper extends the existing MBound algorithm to probabilistically count the number of linear regions, with experiments (Figure 1) demonstrating the speed of this lower bound algorithm compared to counting.

Clarity: The presentation for this paper is relatively clear, but it is quite technical, so some parts are hard to follow, without knowing the prior work in detail.

Originality: Defining A_l(k) and I_l(k) for a refined upper bound, as well as the idea of using a probabilistic lower bound is new compared to prior work.

Comments on Quality and Significance:

The theoretical results presented in this paper are interesting and novel, both the bounds and the adaptation of existing methods (Nemhauser 1978; Gomes 2006) for purposes of estimating bounds. However, I'm uncertain as to the practical applications. One thing that was unclear to me was what Proposition 3, 4 mean for the quantities A_l(k) and I_l(k) in practice (in trained networks). The text makes a comment on the weights and biases having the same number of positive/negative elements but that is likely to only be true for random networks.  It would be interesting to see Figure 1 left for random and trained networks.

Given the long line of work in this area however, I think this paper will be interesting to the community.

---

> ### Author Response · Authors · 2018-11-15
> **Answer to AnonReviewer3**
>
> We appreciate your careful reading and appreciation of the paper. We have made some changes to the manuscript based on your input and addressed some of your comments below.
>
> 1) Clarity
>
> We have rewritten how we pose the problem with current approaches later in the introduction:
>
> “Although the number of linear regions has been long conjectured and recently shown to work for comparing similar networks, this metric would only be used in practice if we come up with faster methods to count or reasonably approximate such number. Our approach in this paper consists of introducing empirical upper and lower bounds, both of which based on the weight and bias coefficients of the networks, and thus able to compare networks having the same configuration of layers.”
>
> Besides the change above, we expanded the introductory paragraph of Section 4 and we included some explanations at the beginning of each subsection to help the reading.
>
> Section 4:
>
> “We prove a tighter – and empirical – upper bound in this section. This bound is obtained by taking into account which units are stably active and stably inactive on the input domain X and also how many among the unstable units are locally stable in some of the linear regions. Before discussing this bound in Section 4.2 and how to compute its parameters in Section 4.3, we discuss in Section 4.1 other factors that have been found to affect such bounds in prior work on this topic.”
>
> Section 4.1:
>
> “The two main building blocks to bound the number of linear regions are activation hyperplanes and the theory of hyperplane arrangements. We explain their use in prior work in this Section.”
>
> Section 4.2:
>
> “Now we show that we can further improve on the sequence of bounds previously found in the literature by leveraging the local and global stability of units of a trained network, which can be particularly useful to compare networks having the same configuration of layers.”
>
> Section 4.3:
>
> “Finally, we discuss how the parameters introduced with the empirical bound in Section 4.2 can be computed exactly, or else approximated.”
>
>
> 2) About practical use
>
> In the long run, we believe that this line of work may help defining necessary and sufficient conditions for different neural networks to be equivalent. However, we also note that recent work in this topic has found some relation between network accuracy and the number of linear regions (Serra et al., 2018), although computing this number exactly is very expensive. We hope that our methods facilitate using the number of linear regions when analyzing neural networks.
>
>
> 3) Comments about Section 4.3
>
> Propositions 3 and 4 are based on the preceding discussion, and they give us a guaranteed bound for the two parameters. Because they involve taking the maximum of expressions over a very large family of subsets, we briefly discussed how these values could be easily computed when the number of positive and negative elements are roughly the same. We have included the following comment to clarify that we do not take this assumption as a fact:
>
> “In cases where this assumption does not hold, we discuss later how to approximate I_l(k).”
>
> We believe that it would help the reader if we compared the values of A_l(k) and I_l(k) in trained networks with those obtained with random networks. We will work on that in the next days.

---

> > ### Author Response · Authors · 2018-11-24
> > **Computing A_l(k) and I_l(k)**
> >
> > We computed the maximum values of k needed in practice to maximize A_l(k) and I_l(k). A summary of the results can be found in Table 2 of Appendix F (page 15). In addition, we added the following to Section 6:
> >
> > "Following up on the discussion from Section 4.3 about computing the values of A_l(k) and I_l(k), we report in Table 2 of Appendix F the minimum value of k to find the maximum value of both expressions for layers 2 and 3 of the trained networks. We observe in practice that such values of k remain small and so does the number of subsets of units from layer l-1 that we need to inspect."
> >
> > In cases where the values of k become larger and prohibitive, the submodular heuristics can be helpful nonetheless.

---

### Official Review · AnonReviewer1 · 2018-11-07
**A good paper, could be improved**

**Rating:** 6
**Confidence:** 4

**Review:**

This paper contributes to the study of the number of linear regions in ReLU neural networks. An approximate probabilistic counting algorithm is used to estimate the lower bound of that quantity, whereas an upper bound is derived analytically. The probabilistic counting algorithm is shown to be much more efficient than exact counting, and is adapted from the SAT literature. The new upper bound uses the weights of the network, a new technique compared to previous work on these bounds, and is shown to be sometimes tighter than the older bound.

Overall, I am positive about the paper. Although I could not verify all proofs in detail, the ones I did verify were sound. The probabilistic counting algorithm seems like a good fit for this type of neural network problems, and is adapted and implemented nicely.

In my opinion, the paper can be improved substantially on these fronts:
- Motivation: Can you point me to a reference where the number of linear regions is used as a measure of expressiveness, formally? I ask because the scope of the work in this paper is very much tied to that question.

- Clarity: this issue must be addressed. The paper is quite technical (that's fine), but also difficult to parse. For example, it is not clear what's new in 4.1.

Minor:
- Figure1/Table1: please move them to experiments. You do not describe the tables and results early, which makes it useless at that stage of the paper. Why not just move them to experiments and describe/discuss these results in detail there?
- Notation: In page 3, paragraph 2, you use x in many different shapes and forms (e.g. bold). Please consider making that notation consistent.

---

> ### Author Response · Authors · 2018-11-15
> **Answer to AnonReviewer1**
>
> Thank you for your time and careful evaluation of our results. We made some changes to the manuscript, and we point below how they reflect your comments and suggestions:
>
> 1) Motivation
>
> Indeed, the number of linear regions has been discussed for a long time, but only recently there were some empirical results showing that such number can be related to the number of linear regions. We included one extra sentence to the first paragraph of the introduction:
>
> “The theoretical analysis of the number of input regions in deep learning dates back to at least Bengio (2009), and more recently Serra et al. (2018) has shown empirical evidence that the accuracy of similar sized rectifier networks is related to the number of linear regions.”
>
> We have also rewritten how we pose the problem with current approaches later in the introduction:
>
> “Although the number of linear regions has been long conjectured and recently shown to work for comparing similar networks, this metric would only be used in practice if we come up with faster methods to count or reasonably approximate such number. Our approach in this paper consists of introducing empirical upper and lower bounds, both of which based on the weight and bias coefficients of the networks, and thus able to compare networks having the same configuration of layers.”
>
>
> 2) Clarity
>
> Besides the changes above, we expanded the introductory paragraph of Section 4 and we included some explanations at the beginning of each subsection to help the reading.
>
> Section 4:
>
> “We prove a tighter – and empirical – upper bound in this section. This bound is obtained by taking into account which units are stably active and stably inactive on the input domain X and also how many among the unstable units are locally stable in some of the linear regions. Prior to discussing this bound in Section 4.2 and how to compute its parameters in Section 4.3, we discuss in Section 4.1 other factors that have been found to affect such bounds in prior work on this topic.”
>
> Section 4.1:
>
> “The two main building blocks to bound the number of linear regions are activation hyperplanes and the theory of hyperplane arrangements. This section talks about the prior work on these two building blocks.”
>
> Section 4.2:
>
> “Now we show that we can further improve on the sequence of bounds previously found in the literature by leveraging the local and global stability of units of a trained network, which can be particularly useful to compare networks having the same configuration of layers.”
>
> Section 4.3:
>
> “Finally, we discuss how the parameters introduced with the empirical bound in Section 4.2 can be computed exactly, or else approximated.”
>
>
> 3) Figure and Table
>
> We agree with your suggestion, and we have moved both to the appropriate section.
>
>
> 4) Notation
>
> We revised the formulations to follow the mathematical conventions regarding matrices, vectors, and sets.

---

### Author Response · Authors · 2018-11-15
**Comments to reviewers and AC**

We would like to thank all reviewers and the AC for the feedback. We have made some preliminary changes to the presentation of the paper to make it more accessible and to address some of the questions raised in the reviews. One of the main concerns is about the practical significance of linear regions. Serra et al. (2018) have shown that the accuracy of similar sized rectifier networks is related to the number of linear regions. We believe that this metric could be helpful for other ends, such as model compression and defense against adversarial perturbation. However, we first need faster algorithms to count or approximate the number of linear regions. This paper addresses this problem. We welcome any other comments you may have.

---

### Meta-Review · Area_Chair1 · 2018-12-16
**Developments on counting linear regions, applicability uncertain**

**Confidence:** 4
**Recommendation:** Reject

**Metareview:**

The paper seeks to obtain faster means to count or approximately count of the number of linear regions of a neural network. The paper improves bounds and makes an interesting contribution to a long line of work.

A consistent concern of the reviewers is the limited applicability of the method. The empirical evaluation can serve to better assess the accuracy of theoretical bounds that have been obtained in previous works, but the practical utility is not as clear yet.

This is a borderline case. The reviewers lean towards a positive rating of the paper, but are not particularly enthusiastic about the paper. The paper makes good contributions, but is just not convincing enough.

I think that the work program that the authors suggest in their responses could lead to a stronger paper in the future. In particular, the exploration of necessary and sufficient conditions for different neural networks to be equivalent and the use of number of linear regions when analyzing neural networks, seem to be very promising directions.